# Statistical Comparison of the Mechanical Properties of 3D-Printed Resin through Triple-Jetting Technology and Conventional PMMA in Orthodontic Occlusal Splint Manufacturing

**DOI:** 10.3390/biomedicines11082155

**Published:** 2023-07-31

**Authors:** Ioan Barbur, Horia Opris, Bogdan Crisan, Stanca Cuc, Horatiu Alexandru Colosi, Mihaela Baciut, Daiana Opris, Doina Prodan, Marioara Moldovan, Liana Crisan, Cristian Dinu, Grigore Baciut

**Affiliations:** 1Department of Maxillofacial Surgery and Implantology, Iuliu Hatieganu University of Medicine and Pharmacy, 400012 Cluj-Napoca, Romania; drbarur@gmail.com (I.B.); crisan.bogdan@umfcluj.ro (B.C.); mbaciut@umfcluj.ro (M.B.); daiana.opris@umfcluj.ro (D.O.); lcrisan@umfcluj.ro (L.C.); cristian.dinu@umfcluj.ro (C.D.); gbaciut@umfcluj.ro (G.B.); 2Department of Polymer Composites, Institute of Chemistry Raluca Ripan, Babes-Bolyai University, 400294 Cluj-Napoca, Romania; stanca.boboia@ubbcluj.ro (S.C.); doina.prodan@ubbcluj.ro (D.P.); marioara.moldovan@ubbcluj.ro (M.M.); 3Department of Medical Education, Division of Medical Informatics and Biostatistics, Iuliu Hatieganu University of Medicine and Pharmacy, 400012 Cluj-Napoca, Romania

**Keywords:** polymethylmethacrylate (PMMA), 3D printing, polyjet, mechanical properties, occlusal splint, scanning electron microscopy (SEM)

## Abstract

Dental 3D-printing technologies, including stereolithography (SLA), polyjet (triple-jetting technology), and fusion deposition modeling, have revolutionized the field of orthodontic occlusal splint manufacturing. Three-dimensional printing is now currently used in many dental fields, such as restorative dentistry, prosthodontics, implantology, and orthodontics. This study aimed to assess the mechanical properties of 3D-printed materials and compare them with the conventional polymethylmethacrylate (PMMA). Compression, flexural, and tensile properties were evaluated and compared between PMMA samples (*n* = 20) created using the “salt and pepper” technique and digitally designed 3D-printed samples (*n* = 20). The samples were subjected to scanning electron microscope analysis. Statistical analysis revealed that the control material (PMMA) exhibited a significantly higher Young’s modulus of compression and tensile strength (*p* < 0.05). In the flexural tests, the control samples demonstrated superior load at break results (*p* < 0.05). However, the 3D-printed samples exhibited significantly higher maximum bending stress at maximum load (MPa) (*p* < 0.05). Young’s modulus of tensile testing (MPa) was statistically significant higher for the control samples, while the 3D-printed samples demonstrated significantly higher values for elongation at break (*p* < 0.05). These findings indicate that 3D-printed materials are a promising alternative that can be effectively utilized in clinical practice, potentially replacing traditional heat-cured resin in various applications.

## 1. Introduction

The use of occlusal devices is regarded as an evidence-based therapeutic option for the management of temporomandibular disorder symptoms [1]. Bruxism, myalgia of the temporomandibular muscles, and arthralgia of the temporomandibular joints are among the indications for using occlusal splints. The materials used to make such devices should be strong enough to bear significant occlusal forces [2]. Typically, acrylic resin that has been auto-, heat-, or light-polymerized is used to make conventional occlusal devices [3]. Recently, additive (3D-printing) techniques have been made possible by manufacturing methods assisted by CAD-CAM technology [4].

Dental 3D printers employ stereolithography (SLA), polyjet (triple-jetting technology), fusion deposition modeling printing, digital light processing, and selective laser sintering [4,5]. SLA printing uses ultraviolet lasers to sculpt resin. In this method, the printing plate travels down in small increments, and the liquid polymer is subjected to an ultraviolet laser that cures a cross section layer by layer. This process is repeated until a dental model is produced [6].

Polyjet 3D printing is comparable to inkjet printing, except that the printer sprays layers of curable liquid photopolymer onto a construction platform [7]. More material is placed immediately on the preceding layer when the construction platform goes down one layer. This is repeated until the form is printed [8]. Triple-jetting technology allows 3D printing of complex things with different materials. Triple jetting uses three printheads to extrude various materials simultaneously, unlike single-nozzle 3D printing. This new method deposits model, support, and contrasting materials precisely and simultaneously. The printheads layer materials to generate intricate shapes and geometries. This approach is useful for prototyping, functioning pieces, and designs that require a mix of materials for mechanical, aesthetic, or functional features. The triple-jetting method allows for more flexibility and efficiency in 3D printing personalized goods [9].

A powerful laser is used in the additive manufacturing technique known as SLS (Selective Laser Sintering) to selectively fuse powdered materials, layer by layer, to produce three-dimensional things out of a variety of materials, including plastics, metals, and ceramics [5].

Recent mechanical property studies of printable denture-based resins have been helpful. The PMMA samples had the highest mean surface roughness, Vickers hardness number, and flexural strength but the lowest contact angle. The samples printed using the ASIGA resin had the highest average contact angle and smoothest average surface, while the Dentona 3D printer had the highest mean impact force. NextDent materials had the highest mean bending modulus and the lowest mean Vickers hardness number, flexural strength, and impact strength [10].

In a separate study on 3D-printed materials, the printed group had the lowest roughness before polishing, the lowest bacterial adherence after 90 min, and superior flexural qualities except for strength. Polished PMMAs had similar surface roughness and microbial adherence [11].

Flexural testing showed that neither the IvoBase printed group nor Vertex ThermoSens specimens fractured under load, while other groups had flexural strength ratings of 71.7 ± 7.4 MPa to 111.9 ± 4.3 MPa. Flexural strength and surface hardness varied from 67.13 ± 10.64 MPa to 145.66 ± 2.22 MPa, indicating significant differences between the tested materials. CAD/CAM and polyamide had the highest flexural strength, while a third group and polyamide had the lowest. Flexural strength was lowest in 3D-printed materials [12].

Antibacterial 3D-printed materials have been enhanced using various methods. Stereolithography-printed aluminum nitride composites had good dispersion and antibacterial properties, reducing colony-forming units by 70%. Aluminum nitride-reinforced PMMA resins also had good mechanical properties, with a 12 percent loss of ultimate strength for ceramic fractions of 15 percent and the potential for further strengthening through conventional post-curing [13].

Three-dimensional-printed materials may have lower mechanical properties than heat-cured PMMA, according to some studies. ZrO_2_NPs in 3D-printed resins increased flexure strength, impact strength, and hardness (*p* < 0.05) but not surface roughness or elastic modulus. Three-dimensional-printed resins with ZrO_2_NPs had better mechanical properties than heat-polymerized acrylic resin. The new nano-composite denture-based resins may be clinically applicable [14].

Considering these advancements, the present study aimed to investigate the mechanical properties of 3D-printing materials and compare them with the conventional polymethylmethacrylate. Specifically, we evaluated and compared the compression, flexural, and tensile properties of samples manufactured using these two approaches. Additionally, scanning electron microscopy was employed to examine the surface characteristics of the investigated samples. By elucidating the mechanical performance of 3D-printed occlusal splints, this research contributes to our understanding of their potential applications in clinical practice.

## 2. Materials and Methods

### 2.1. Manufacturing of Experimental Samples

#### 2.1.1. PMMA

For each mechanical trial, 20 samples were created using polymethylmethacrylate (Orthocryl^®^—Dentaurum, Ispringen, Germany) and the “salt and pepper” technique in accordance with the manufacturer’s instructions.

According to the manufacturer’s instructions, the salt-and-pepper method begins by first combining salt and pepper. Applying a coating of powder (polymer), followed by a layer of liquid (monomer), and then applying alternating layers of powder and liquid with tiny pendulum motions. Apply only as much liquid as the powder can absorb. The substance must not run off. To prevent air pockets from forming under the screws, a slightly larger amount of fluid at the outset can be applied so that the acrylic can spread under the screws. Powder must be used to complete the application. Sufficient powder must be applied to dry the top layer.

The samples were placed for curing in the pressure vessel containing water at between 40 and 46 °C/104 and 115 °F and a pressure of 2.2 bars (30 p.s.i.) for 20 min. The samples included flat samples for tensile testing, parallelepipedal prisms for flexing (2 mm × 2 mm × 25 mm), and standard cylinders (4 mm diameter × 8 mm length) for compression.

#### 2.1.2. 3-D Printed

Using dental CAD software program (Exocad; Exocad GmbH, Darmstadt, Germany), the samples were digitally designed, and with the help of the 3D printer (Stratasys Ltd., Eden Prairie, MN, USA), the samples were printed from biocompatible clear resin MED610 (Stratasys Ltd., Eden Prairie, MN, USA) according to specifications of the manufacturer. Twenty pieces were produced with the same dimensions as the control samples, which were manufactured from PMMA.

### 2.2. Evaluation of Mechanical Properties

For the mechanical evaluation, the samples were produced in conformity with ISO 4049:2019 [6]. Throughout the preparation process, silicone molds were used to lessen the formation of cracks and flaws within the material.

A universal testing device (LR5K Plus, Lloyd Instruments Ltd., West Sussex, UK) with a maximum allowable capacity of 5 KN was used to test the mechanical characteristics at room temperature of 23 °C. Data processing was carried out using Nexygen software (version 4.0) [15]. According to ISO 4049:2019 [6], the samples (*n* = 20) were polymerized at dimensions under the manufacturer’s conditions.

Three various mechanical tests were conducted:The compressive strength test (Figure 1), which gauges the material’s stiffness and determines the breaking point and compressive strength curve. The Nexygen software generated test results while each compression sample was being tested at a speed of 0.5 mm/min and a force of 50 N/s. The samples used for this test were cylindrical in form, measuring 6 mm in height and 4 mm in diameter.

2.The three-point bending test was used to conduct the bending resistance test (Figure 2), which measured bending resistance and Young’s modulus. The samples were rectangular prisms with dimensions of 25 mm in length, 2 mm in height, and 2 mm in width. With a force of 50 N/min, the arm was lowered at a pace of 1 mm/min.

3.To determine the tensile strength (Figure 3), a test piece was subjected to an increasing axial force, typically until it broke, while the accompanying changes in the test piece’s length were recorded. Based on the ASTM-D638 reference standard [16], measurements were performed using a load force of 5 N and a stress of 1000 N/mm^2^. The test pieces utilized for the tensile test were flat rectangular shapes with dimensions of 2 mm thick, 6 mm wide, and 40 mm long. They provided the testing equipment with a calibrated part and two gripping ends (2 mm × 10 mm × 10 mm).

### 2.3. Scanning Electron Microscopy

Scanning electron microscopy was performed on the fillers and the samples using an FEI Inspect microscope (SEM-Inspect S, FEI Company, Hillsboro, OR, USA), S model, functional in high-vacuum and low-vacuum, with an accelerating voltage between 200 V and 30 kV (discs). The microscope had a 4096 × 3536 pixel image-processing capability CCD-IR infrared inspection camera in addition to a backscatter electron detector. The images were typically taken at magnifications of 500–1000 times.

### 2.4. Statistical Analysis

JASP Version 0.16.3 (JASP Team, University of Amsterdam, Amsterdam, The Netherlands) [17] has been used to statistically describe and analyze the measured mechanical characteristics.

Shapiro–Wilk tests and Q–Q plots have been used to assess the normality of data distributions resulting from the measured characteristics.

Descriptive statistics have been computed and reported. Median values and interquartile ranges (IQR) were considered reliable measures of central tendency and data spread, respectively, regardless of the symmetrical or skewed distribution of the described data. Mean values and standard deviations (SD) were also reported as indicators of central tendency and data spread, with a higher reliability in the case of normally distributed data (*p* > 0.05 following the Shapiro–Wilk test). 

Independent samples *t*-test was used to evaluate the degree of significance between the samples. The level of statistical significance was chosen to be 0.05 for all comparisons.

The mechanical characteristics of the examined materials were graphically presented and contrasted using boxplots, raincloud plots, and plots for means and their 95% CI.

## 3. Results

### 3.1. Compression

In Table 1, the results of the Young’s modulus of compression and tensile strength are represented. Only valid registrations of the tested samples have been used in this analysis. The values for the Young’s modulus of compression for both printed and control samples have a normal distribution (*p* < 0.05). The tensile strength for printed samples also has a normal distribution (*p* < 0.05), and the control has a non-normal distribution.

In Figure 4, the aspect of the samples after the mechanical compression test can be observed with the macroscopic structural modification accordingly.

In Figure 5, the raincloud plots for the Young’s modulus of compression and tensile strength can be observed. The 3D-printed material has lower median values for both tests. It can also be observed that the 3D-printed material has a smaller interquartile distance and a smaller variability in the samples tested.

In Table 2, the results of the independent samples *t*-test for Young’s modulus of compression and tensile strength can be observed with statistically significant values for both tests *p* < 0.05. This results in the rejection of the null hypothesis that there is no significant difference between the samples tested, meaning that there is a statistically significant difference in the compression test between the 3D-printed material and the standard PMMA.

### 3.2. Bending/Flexural

In Figure 6, the aspect after the flexural mechanical test can be observed with the maximum bend for the tested sample.

In Table 3, the descriptive statistics can be observed for the mechanical bending/flexural tests: Young’s modulus of bending, load at break, maximum bending stress at maximum load, and stiffness of the investigated materials (control PMMA and 3D printed). The number of valid tests can be observed. There is normal distribution (*p* < 0.05) for the Young’s modulus of bending (PMMA and printed) and for stiffness (both samples).

In Figure 7, the raincloud plots for the bending mechanical test can be observed. For the Young’s modulus of bending, we can observe that there are a few outliers for the control (PMMA). The load at break median is higher for the printed material with a lower interquartile distance and a lower variability. The maximum bending stress at maximum load is lower for the 3D-printed material with a tighter interquartile distance. The stiffness raincloud plot shows us that there is a high variance in the PMMA samples.

Table 4 presents the independent samples *t*-test for the mechanical bending tests for the PMMA and 3D-printed material. The 3D-printed material has higher load at break (*p* < 0.001), and the PMMA has a higher maximum bending stress at maximum load (*p* < 0.001).

### 3.3. Tensile Testing

In Figure 8, the aspect after the mechanical tensile test can be observed.

Table 5 presents the descriptive statistics for the mechanical tensile test for 3D printed and the control PMMA Young’s modulus of tensile testing and elongation at break with the valid samples. The Young’s modulus of tensile testing has a normal distribution (*p* < 0.05) for both samples, and the 3D printed has a normal distribution for the elongation at break (*p* < 0.05).

In Figure 9, the raincloud plots for the mechanical tensile testing can be observed. The median for Young’s modulus of tensile testing is lower for the 3D printed, and the control has a lower elongation at break. These values have a significant statistical difference, as shown in Table 6.

### 3.4. SEM Analysis

Scanning Electron Microscopy (SEM) (Figure 10 and Figure 11) was used to analyze the morphology of the samples, with low magnification being used for the observation of major morphologic features and a microstructure overview, medium magnification being used for precise morphological features, and high magnification being used for microstructural details.

In Figure 10, the PMMA sample in this study has a compact microstructure generated by PMMA particles diffusing toward one another, forming strong connection necks due to the photopolymerization process. At 500× magnification, many polyhedral pores are visible, indicating the effectiveness of the consolidation process. The material diffusion from left to right is evident in the upper left side, with a tendency for pores to form. SEM microscopy was used to analyze the fracture microstructure following the flexural strength test, showing complex solicitations such as lateral compression and extension. The failure of the PMMA sample is evident on the lower side, with interior pore rupture and PMMA neck failure close to the surface.

The compression determines the densification of the material; the holes decrease, increasing overall resistance. The failure spreads through the material until it reaches the dense layer, which breaks the sample. The printed PMMA (Figure 11) sample was generated layer by layer, with a mean layer thickness of 350 µm and a transition zone of 50 µm. The flexural strength of the material was tested perpendicular to the inter-layer transition zones, and the fracture aspect is complex depending on microstructural aspects. The compressed area on the lower side of the image shows good preservation of microstructural aspects due to compression effort, and failure occurs simultaneously in both layers and transition zones.

## 4. Discussion

Regarding the compression, our results concluded that the control material (PMMA) has a statistically significant higher Young’s modulus of compression and tensile strength (*p* < 0.05). Concerning the flexural tests, statistically significant and higher results were shown to be better for the control samples (PMMA) for load at break. Maximum bending stress at maximum load (MPa) is statistically more significant for the 3D-printed samples. Young’s modulus of tensile testing (MPa) is statistically significant for the control samples. The 3D-printed samples had statistically significant values for elongation at break.

The results of our study cannot clearly conclude which of the material behaves better, as the control behaves better in some conditions and the 3D printed using the polyjet technique in other conditions. The polyjet technique has brought some improvements in the splint quality, and it also inherently has other advantages, such as reproducibility of the technique, fast learning curve for the technician, scalability, less residual material and lower toxicity, uniformity of the product, and mechanical proprieties that are the same in all the thicknesses of the splint.

A review of the literature has shown similar results regarding the mechanical features of the polyjet materials having better flexural strength [18].

To summarize the results of our study, PMMA shows better compression results with higher values, whereas the 3D-printed polyjet material presents better tensile and flexural mechanical results. This anisotropy is in accordance with other studies, which find that modifying the force direction majorly influences the outcome of the mechanical testing [19].

Additionally, considering the technical aspect of the 3D-printing polyjet technique, the mechanical proprieties differ due to the placement and orientation on the XYZ on the printing tray, according to some studies [20]. Some papers suggest that the tensile proprieties of the material, when printed along the Z-axis, are improved [21].

Other parameters, such as printing mode and type of finish, can be modified in the printing process to enhance some proprieties, such as the tensile strength [22]. Additionally, postprocessing of polishing can also significantly modify the results of the mechanical testing [23].

Figure 10 shows a compact microstructure generated by PMMA particles diffusing toward one another to form strong connection necks because of the photopolymerization process in the PMMA sample prior to the flexural test. However, at 500× magnification, many pores are seen that are polyhedral in shape and range in size from 80 to 200 nm. In Figure 10, a pore detail shown at 1000× magnification shows the strong bonding of the photopolymerized PMMA particles. The formation of tiny PMMA filaments on the pores’ surface is evidence of the consolidation process’ effectiveness. The neck between two previous PMMA particles can be seen in the middle-lower half of the image at a high magnification level of 5000×, demonstrating the material diffusion from left to right. The image’s upper left side demonstrates a tendency for pores to form. Because of the energy being dissipated through the pore walls, which enhances the material tenacity, this complex structure may be exceedingly resilient.

SEM microscopy was also used to analyze the fracture microstructure following the flexural strength test. The results are shown in Figure 10. Complex solicitations, such as lateral compression on the upper side and extension on the lower side of the testing specimen, take place. The energy is lost through the pores and produces intricate micro solicitations over the PMMA necks inside the substance. Due to the breaking line’s progressive elongation observable at low magnification, the failure first manifests itself on the lower side. Average magnification demonstrates interior pore rupture and PMMA neck failure close to the surface. It happens when necks elongate past the point of greatest resistance, as seen at extreme magnification (5000×). On the middle-lower side of the image, a broken neck can be seen, with the fracture margin looking like a “bent plastic sheet” due to the intense elongation effort.

On the plus side, the compression determines the densification of material, and the holes decrease, increasing the overall resistance. The failure, which started on the specimen’s lower side, gradually spreads through the material until it reaches the dense layer, which suddenly becomes exposed during the elongation effort and completely breaks the sample, as seen in the right side of the SEM image at low magnification in Figure 10. As a result, the PMMA sample’s overall behavior shows good tenacity but just average flexural strength. The addition of a modest amount of micro-sized filler material may help the situation.

The printed PMMA sample was generated layer-by-layer by adding material that was locally photopolymerized. Thus, the SEM image at low magnification (100×) in Figure 11 presents a mean layer thickness of about 350 µm. Successive layers are bonded together by a transition zone of about 50 µm. Moderate magnification (×500) reveals a compact structure of the PMMA inside of the printed layers having a relative dendritic aspect and a low number of pores. Only a few rounded pores are observed ranging from 10 to 30 µm in diameter. This compact structure inside of the printed layers assures good cohesion of the material. Unfortunately, the high magnification SEM images (1000× and 5000×) in Figure 11 indicate a fine-grained structure of the layer transition zone, which is a sign of a lower cohesion due to the partial lack of photopolymerization. The presence of small, rounded PMMA grains of about 5 µm might be sensitive to certain effort dissipation through the material.

The flexural strength of the material was tested perpendicular to the inter-layer transition zones, as observed at low magnification (100×) in Figure 11, and the fracture aspect is complex depending on the microstructural aspects. Therefore, the average magnification (500×) in Figure 11 reveals the lower part of the sample that supports the elongation effort in the upper side of the image. The transition zones are very tensioned and elongated, proving their behavior as failure promoters, circumstances better observed at higher magnifications (1000× and 5000×). Since they disintegrated, the failure is further propagated into the printed layers. The compressed area situated on the lower side of the image reveals good preservation of the microstructural aspects due to the compression effort, and the failure occurs simultaneously in both layers and transition zones.

SEM investigation proves that lower values obtained for the flexural strength of printed PMMA compared to the bulk samples are generated by the behavior of the transition zones between printed layers. There might be an improvement in the quality of the material with more control of the parameters allowing a better adhesion between printed layers.

### Limitations

This study was conducted with some constraints regarding the sample size and the number of products to be compared. Ideally, without any space and physical constraints, more samples in different forms should be used.

Only one type of printing direction was used, and this can be a limitation due to the increased anisotropy of the mechanical proprieties in such cases. Additionally, no tweaking of other proprieties of the printing process was performed, such as surface finish or speed of printing.

Assessing materials with the purpose of dental use must consider the oral environment, which is complex. Aging the materials modifies the proprieties and can drastically change the proprieties and the behavior for the foreseen period of use of the appliance [24].

Only one type of printer and one type of material were used.

Some studies suggest that there is no statistical significance in different 3D-printed materials using another type of printing method (low-force stereolithography) when assessing mechanical proprieties [25,26].

Further studies should, therefore, be performed to investigate whether triple-jetting technology could be suitable for occlusal retainers after conventional [27], lingual [28], or aligner-based orthodontic treatments [29]. Additionally, adding different nanofillers [30] to the compound could enhance some proprieties, as seen in previous studies [10,11,12,13,14,31,32].

## 5. Conclusions

The findings of this study demonstrate that 3D-printed materials, particularly those produced using triple-jetting technology, offer a promising and viable solution for clinical applications. Our research revealed favorable mechanical properties of the 3D-printed material, indicating its potential to replace the conventional heat-cured resin (PMMA) in various applications.

However, further exploration and understanding of 3D-printing protocols and processes are warranted to enhance the mechanical properties of the printed materials. Improvements in these areas will contribute to the optimization of 3D-printed occlusal splints for clinical use.

Additionally, while polyjet technology is still under development, future research endeavors can expand its capabilities and explore its potential in the field of orthodontic occlusal splint manufacturing. The ease of use and comparable mechanical properties of polyjet technology make it an appealing avenue for further investigation.

In conclusion, the present study highlights the significant potential of 3D-printed materials in clinical practice. PMMA showed better compression results with higher values, whereas the 3D-printed polyjet material presented better tensile and flexural mechanical results. However, ongoing research and advancements in 3D-printing protocols and technologies are crucial to unlocking the full range of benefits and improving the mechanical properties of these materials. By capitalizing on these developments, the field of orthodontics can continue to progress towards more effective and efficient occlusal splint manufacturing techniques.

## Figures and Tables

**Figure 1 biomedicines-11-02155-f001:**
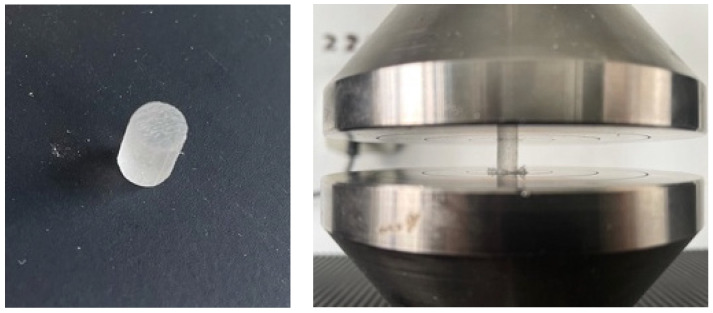
Testing the compressive strength on PMMA cylindric samples (PMMA and 3D-printed sample).

**Figure 2 biomedicines-11-02155-f002:**
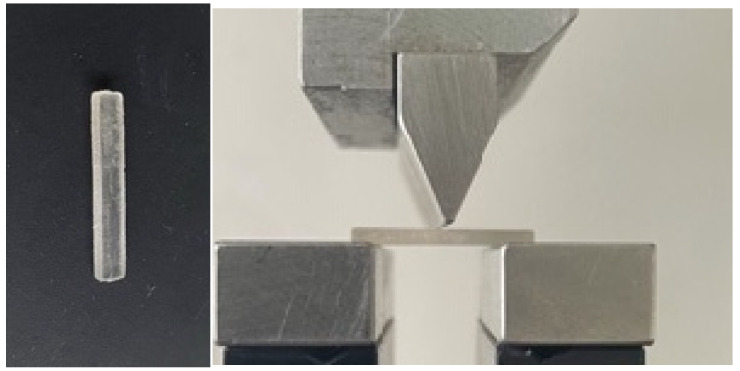
Prior to and throughout the evaluation, rectangular PMMA prisms were used to measure the three-point bending resistance.

**Figure 3 biomedicines-11-02155-f003:**
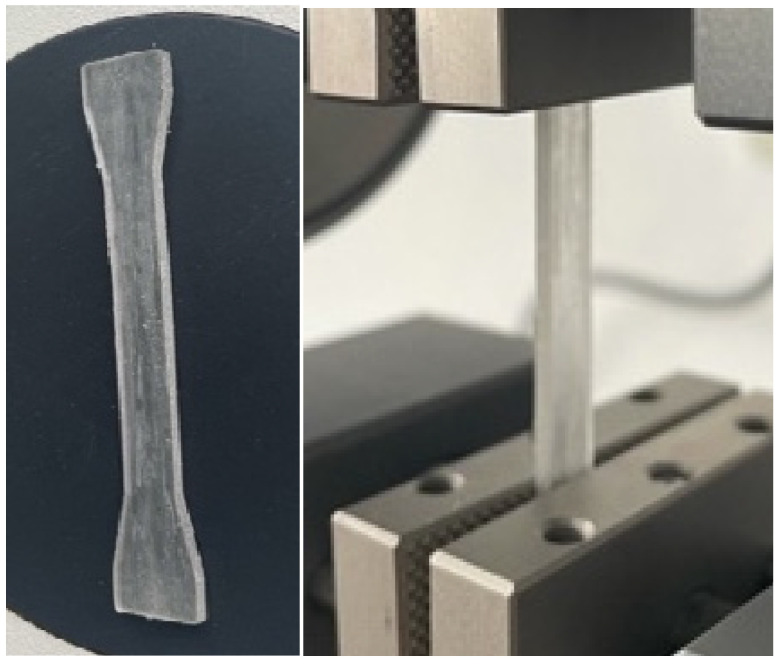
Specimens that were utilized to evaluate the tensile strength prior to and during the evaluation.

**Figure 4 biomedicines-11-02155-f004:**
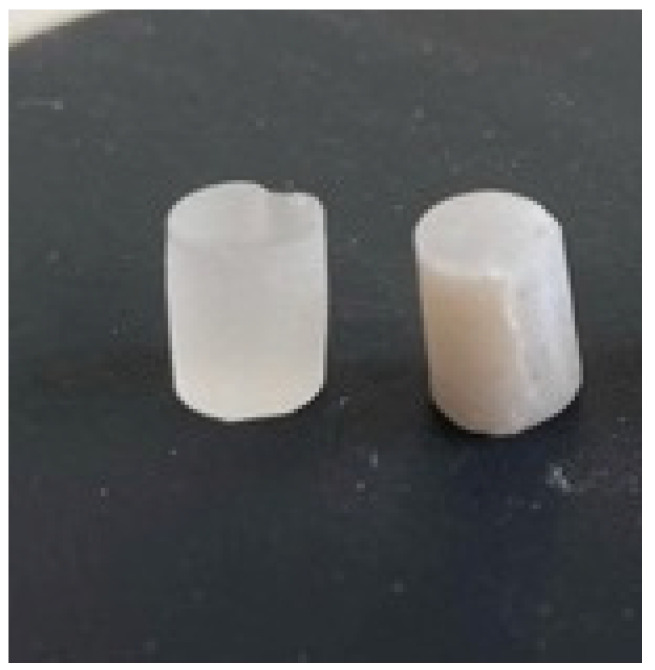
The aspect of the samples after the compression test (left—PMMA control, right—3D printed).

**Figure 5 biomedicines-11-02155-f005:**
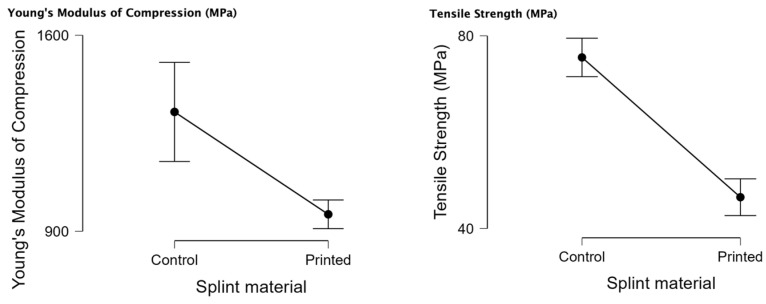
Young’s modulus of compression and tensile strength rain cloud charts and descriptive plots for PMMA (control) and 3D-printed materials.

**Figure 6 biomedicines-11-02155-f006:**
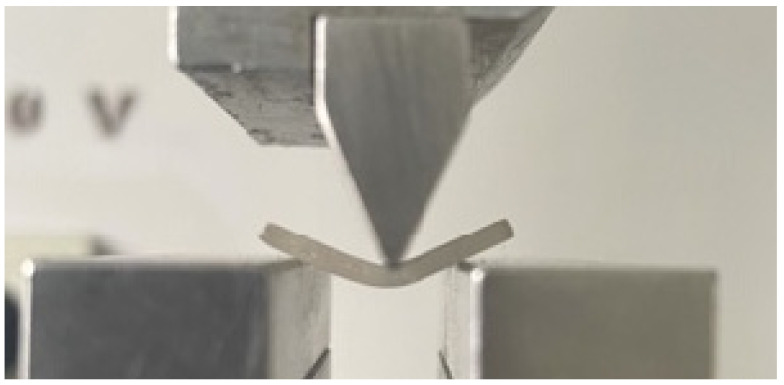
Aspect after the bending/flexural testing of the samples.

**Figure 7 biomedicines-11-02155-f007:**
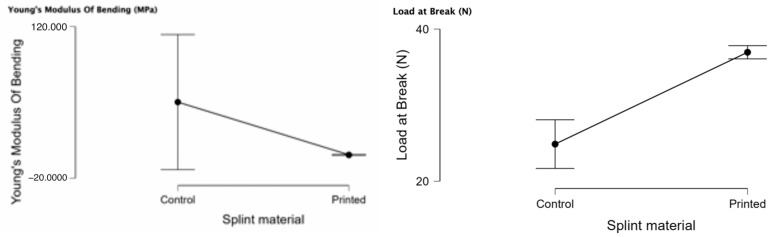
Young’s modulus of bending, load at break, maximum bending stress at maximum load, and stiffness for PMMA (control) and 3D printing, presented as rain cloud plots.

**Figure 8 biomedicines-11-02155-f008:**
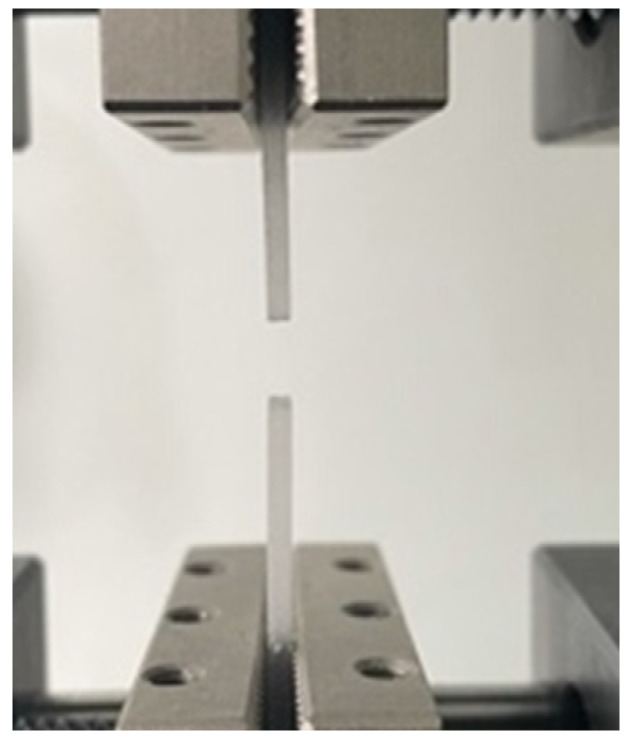
Aspect after the mechanical tensile test of the samples.

**Figure 9 biomedicines-11-02155-f009:**
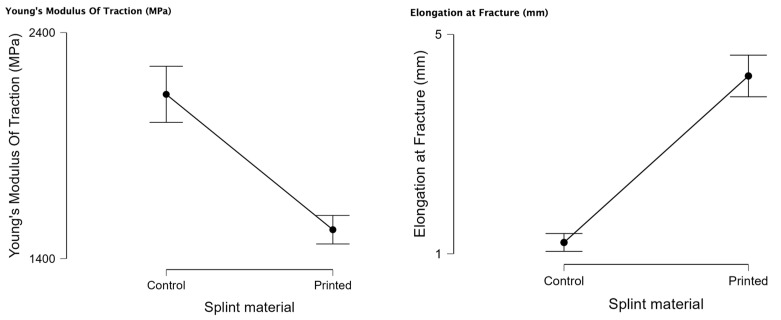
Comparative rain plots for 3D-printed material and PMMA (control) regarding Young’s modulus of tensile testing and elongation at break.

**Figure 10 biomedicines-11-02155-f010:**
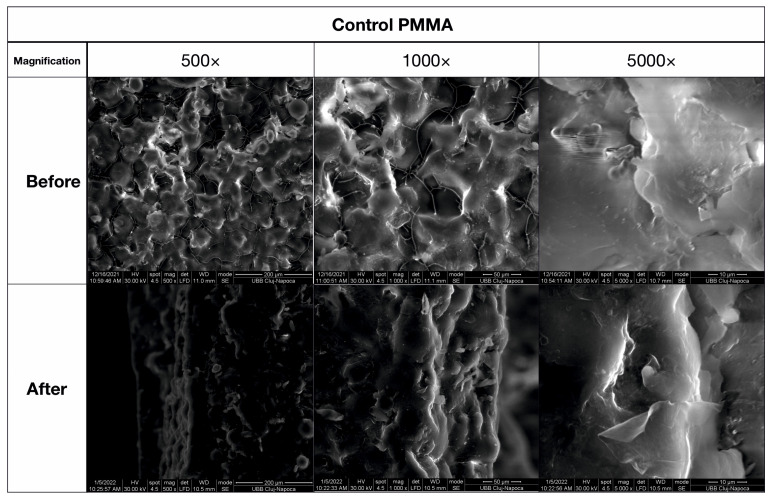
SEM imaging of the PMMA (control) sample before and after flexural strength tests.

**Figure 11 biomedicines-11-02155-f011:**
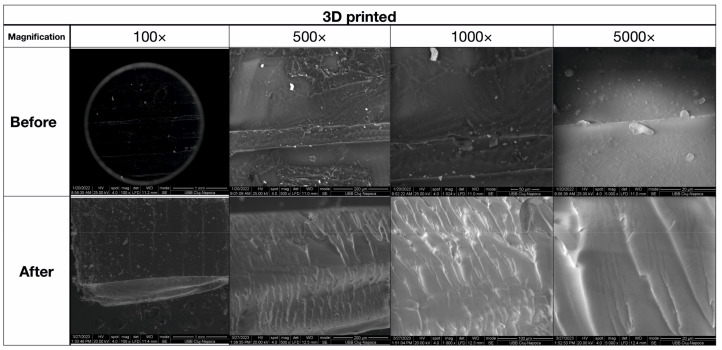
SEM imaging of the 3D-printed sample before and after flexural strength tests.

**Table 1 biomedicines-11-02155-t001:** Young’s modulus of compression and tensile strength: descriptive statistics for PMMA (control) and 3D-printed materials in the compression test (SD—standard deviation; IQR—interquartile range).

	Young’s Modulus of Compression (MPa)		Tensile Strength (MPa)	
	Printed	Control	Printed	Control
Valid N	13	16	13	16
Shapiro–Wilk *p*-value	0.007	<0.001	0.017	0.586
Median	1011.4	1468.61	50.5	76.85
Mean	960.7	1326.18	46.5	75.51
SD	84.8	331.97	6.3	7.49
IQR	167.7	307.19	11.5	10.4
Minimum	853.4	589.36	37.1	63.12
Maximum	1055.7	1598.58	52.9	86.89

**Table 2 biomedicines-11-02155-t002:** Young’s modulus of compression and tensile strength comparison of the materials under investigation (PMMA as control and 3D printed).

	Young’s Modulus of Compression (MPa)	Tensile Strength (MPa)
	Independent Samples *t*-test	
PMMA control vs. 3D printed	0.001	0.001

**Table 3 biomedicines-11-02155-t003:** Young’s modulus of bending, load at break, maximum bending stress at maximum load, and stiffness of the investigated materials (control PMMA and 3D printed). SD—standard deviation; IQR—interquartile range.

	Young’s Modulus of Bending (MPa)		Load at Break (N)		Maximum Bending Stress at Maximum Load (MPa)		Stiffness (N/m)	
	Printed	Control	Printed	Control	Printed	Control	Printed	Control
Valid N	19	23	19	23	19	23	19	23
Shapiro–Wilk *p*-value	0.002	<0.001	0.64	0.174	0.99	0.833	0.002	<0.001
Median	1287.91	7092.49	36.55	21.91	68.1	103.81	53,358	41,222.01
Mean	1443.3	50,237.74	36.95	24.88	68.11	108.04	60,982.5	330,098.91
SD	410.7	144,037.9	1.8	7.4	2.72	29.96	17,286.6	991,464.4
IQR	409.77	7143.04	1.7	11.01	3	44.77	17,372.5	25,595.48
Minimum	1017.5	4115.9	33.6	14.04	62.3	53.94	43,137	25,276.75
Maximum	2394.5	668,077.42	40.9	44.1	74	164.11	101,510	4,575,000

**Table 4 biomedicines-11-02155-t004:** Results of the independent *t*-test for the Young’s modulus of bending, load at break, maximum bending stress at maximum load and stiffness between PMMA (control) and 3D-printed material.

	Young’s Modulus of Bending (MPa)	Load at Break (N)	Maximum Bending Stress at Maximum Load (Mpa)	Stiffness (N/m)
Applied Test	Independent Samples *t*-test			
	*p*-values			
3D printed vs. Control	0.1	<0.001	<0.001	0.2

**Table 5 biomedicines-11-02155-t005:** Young’s modulus of tensile testing and elongation at break: descriptive statistics for PMMA (control) and 3D-printed material in the bending test. SD—standard deviation; IQR—interquartile range.

	Young’s Modulus of Tensile Testing (MPa)		Elongation at Break (mm)	
	Printed	Control	Printed	Control
Valid N	19	22	19	22
Shapiro–Wilk *p*-value	0.035	0.046	0.031	0.397
Median	1496.3	2081.06	4.26	1.23
Mean	1528	2126.85	4.24	1.21
SD	130.7	279.89	0.78	0.37
IQR	166.1	353.65	0.99	0.28
Minimum	1383.2	1795.76	1.9	0.55
Maximum	1879.1	2817.36	5.5	1.96

**Table 6 biomedicines-11-02155-t006:** Comparison of Young’s modulus of tensile testing and elongation at break between the investigated materials (3D printed and control PMMA).

	Young’s Modulus of Tensile Testing (MPa)	Elongation at Break (mm)
	Independent Samples *t*-test	
3D printed vs. Control	<0.001	<0.001

## Data Availability

Not applicable.

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
