# Peer review of "Statistical Comparison of the Mechanical Properties of 3D-Printed Resin through Triple-Jetting Technology and Conventional PMMA in Orthodontic Occlusal Splint Manufacturing"

_biomedicines, 2023, doi:10.3390/biomedicines11082155_

Round 1
Reviewer 1 Report
Please, see the comments in pdf file.
- Title should be rewritten. It should be written clear and concise, not as a sentence.
- Abstract should be rewritten, especially paying attention on technical English expressions. Concept of the abstract should be changed, because there is one “introduction” sentence and then continuing with not well written goal pf research.
o Please, rewrite the first Abstract sentence (lines 18,19).
o After this sentence there should be at least one more introducing the 3D printing and its influence in this research. With that, good "connection" should be made with the rest of the abstract.
- Keywords should be arranged in a different order.
- Lines 41,42 - Reference 3 is not appropriate, concerning the fact that no polymerization is mentioned in the referenced manuscript.
- Lines 43,44 - This sentence is quite confusing. These techniques are completely opposite. Subtractive is conventional, well known (“old”) manufacturing approach, and because of that it is not known from recent period. Additive (3D) printing approach is newer, and in comparison with subtractive it can be said that it appeared recently.
- Line 46 – Reference 4 is also confusing.
- Line 4,48 – This sentence (also in the abstract) should be rewritten. Triple jetting technology is advanced part of polyjet technology, so it should be placed in brackets. Also, not only these techniques are represented in dentistry 3D printing.
- Lines 47-55 - SLA is described, but where are the other two, in previous sentence, mentioned 3D printing techniques applied in dentistry?
- Introduction is too short, it does not give any important, or even relevant information regarding the current situation in the field which authors want to give their scientific contribution. It is not introducing reader into what can be expected on proper manner. 5 references is not enough.
- Figure 1 is not clear et al.
- Line 94 – Three point bending test.
- Please, use regular diagrams for mechanical testing’s which are generated by the testing machine. Not the cloud charts.
- It is necessary to use technical terminology for mechanical testing results.
- All the underlined text in the discussion should be rewritten, re-conceptualized, and placed in the introduction. Leaving almost no discussion.
- Conclusion is weak.
- 14 references in total regarding this subject is too small.

English needs to be completely checked and rewritten. Also, it is necessary to respect the technical writing and terminology, bearing in mind that you are representing the mechanical testing results.
Author Response
Please, see the comments in pdf file.
Point 1:
- Title should be rewritten. It should be written clear and concise, not as a sentence.
Response 1:
Thank you very much for your suggestion. We have modified the title as suggested.
Point 2:
- Abstract should be rewritten, especially paying attention on technical English expressions. Concept of the abstract should be changed, because there is one “introduction” sentence and then continuing with not well written goal pf research.
o Please, rewrite the first Abstract sentence (lines 18,19).
- After this sentence there should be at least one more introducing the 3D printing and its influence in this research. With that, good "connection" should be made with the rest of the abstract.
Response 2:
Thank you very much for your suggestions. We modified the abstract as suggested and we believe that it made it clearer and more concise. Thank you!
Point 3:
- Keywords should be arranged in a different order.
Response 3:
Thank you. We have arranged the keyword to the importance that they have to the article.
Point 4:
- Lines 41,42 - Reference 3 is not appropriate, concerning the fact that no polymerization is mentioned in the referenced manuscript.
Response 4:
Thank you for your remark. We modified the paragraph to be appropriate.
Point 5:
- Lines 43,44 - This sentence is quite confusing. These techniques are completely opposite. Subtractive is conventional, well known (“old”) manufacturing approach, and because of that it is not known from recent period. Additive (3D) printing approach is newer, and in comparison with subtractive it can be said that it appeared recently.
Response 5:
Thank you for your careful reading of our manuscript. We have modified the sentences to be accurate.
Point 6:
- Line 46 – Reference 4 is also confusing.
Response 6:
We modified the reference to better fit the content. Thank you!
Point 7:
- Line 4,48 – This sentence (also in the abstract) should be rewritten. Triple jetting technology is advanced part of polyjet technology, so it should be placed in brackets. Also, not only these techniques are represented in dentistry 3D printing.
Response 7:
Thank you for your feedback! We have rearranged the paragraph according to your suggestions. Thank you!
Point 8:
- Lines 47-55 - SLA is described, but where are the other two, in previous sentence, mentioned 3D printing techniques applied in dentistry?
Response 8:
Thank you. We did not want to exhaustively explain all the technologies involved, but just the one which we believe is newer and innovative (polyjet).
Point 9:
- Introduction is too short, it does not give any important, or even relevant information regarding the current situation in the field which authors want to give their scientific contribution. It is not introducing reader into what can be expected on proper manner. 5 references is not enough.
Response 9:
We have rearranged the entire paragraph to better suit the needs of the reader and also added better references.
Point 10:
- Figure 1 is not clear et al.
Response 10:
We rearranged figure 1 to be a bit more clear.
Point 11:
- Line 94 – Three point bending test.
Response 11:
Thank you very much for your observation. We have corrected it accordingly!
Point 12:
- Please, use regular diagrams for mechanical testing’s which are generated by the testing machine. Not the cloud charts.
Response 12:
We have added the descriptive charts as requested. Thank you.
Point 13:
- It is necessary to use technical terminology for mechanical testing results.
Response 13:
Thank you, we modified the terms accordingly.
Point 14:
- All the underlined text in the discussion should be rewritten, re-conceptualized, and placed in the introduction. Leaving almost no discussion.
Response 14:
Thank you very much for the feedback and the suggestions. We rephrased the discussion and moved it to the introduction section.
We have added new discussion with appropriate references.
Point 15:
- Conclusion is weak.
Response 15:
We rephrased the conclusion. Thank you!
Point 16:
- 14 references in total regarding this subject is too small.
Response 16:
We have rearranged the manuscript and enhanced it. We further added important references in both introduction and discussions sections.
Reviewer 2 Report
Although the title is interesting however, the paper needs thorough revision. These are the few major comments apart from the minor comments:
- Abstract should be rewritten clearly. The numerical or p values are missing from the Abstract
- Introduction is too short. Please write more about techniques being used for splints manufacturing
- Figure 1 is very blurred. Improve the dpi
- The major flaw of the paper is just evaluation of baseline readings. the authors have ignored the effect of artificial ageing. The authors should incorporate the findings related to artificial ageing as well
- The authors have mentioned the outcome of hypothesis in the Results part but they forgot to raise the hypothesis in the introduction part
- Why the section 3.4. is so lengthy? The authors should describe the results briefly and then explain and discuss the results in length in the Discussion part.
- Discussion part is weak. It lacks scientific reasoning and justifications.
moderate editing is required
Author Response
Point 1:
Although the title is interesting however, the paper needs thorough revision. These are the few major comments apart from the minor comments:
Response 1:
Thank you very much for your effort and patience to review our work!
Point 2:
- Abstract should be rewritten clearly. The numerical or p values are missing from the Abstract
Response 2:
Thank you very much for the feecback. We added the p values and rephrased the abstract.
Point 3:
- Introduction is too short. Please write more about techniques being used for splints manufacturing
Response 3:
Thank you very much for the suggestions. We added the necessary information in the introduction.
Point 4:
- Figure 1 is very blurred. Improve the dpi
Response 4:
Thank you very much. We modified Figure 1 so it would be a bit more clear.
Point 5:
- The major flaw of the paper is just evaluation of baseline readings. the authors have ignored the effect of artificial ageing. The authors should incorporate the findings related to artificial ageing as well
Response 5:
We added the explanations in the discussion part. Thank you for the suggestion!
Point 6:
- The authors have mentioned the outcome of hypothesis in the Results part but they forgot to raise the hypothesis in the introduction part
Response 6:
We have modified the text accordingly.
Point 7:
- Why the section 3.4. is so lengthy? The authors should describe the results briefly and then explain and discuss the results in length in the Discussion part.
Response 7:
We summarized section 3.4 and moved the explanations to the discussion section, where they belong. Thank you for the suggestions!
Point 8:
- Discussion part is weak. It lacks scientific reasoning and justifications.
Response 8:
We have restructured the discussion section to add more reasoning and justifications.
Reviewer 3 Report
The manuscript presents interesting study. However I found a few major flaws:
1. Authors have to rationale the study and write why this study is novel and important. Please add a paragraph in Introduction before the aim of the study.
2. Authors cited only 14 references (references should have at least 25 items)! Almost half of them are older than 10 years. Authors should remove all references older than 10 years and consider to cite current and reliable literature related to the topic (citation of mentioned literature is voluntarily):
Paradowska-Stolarz A, Malysa A, Mikulewicz M. Comparison of the Compression and Tensile Modulus of Two Chosen Resins Used in Dentistry for 3D Printing. Materials (Basel). 2022 Dec 15;15(24):8956. doi: 10.3390/ma15248956.
Paradowska-Stolarz A, Wezgowiec J, Mikulewicz M. Comparison of Two Chosen 3D Printing Resins Designed for Orthodontic Use: An In Vitro Study. Materials (Basel). 2023 Mar 10;16(6):2237. doi: 10.3390/ma16062237.
Paradowska-Stolarz AM, Wieckiewicz M, Mikulewicz M, et al. Comparison of the tensile modulus of three 3D-printable materials used in dentistry [published online as ahead of print on May 25, 2023]. Dent Med Probl. doi:10.17219/dmp/166070
Paradowska-Stolarz A, Wezgowiec J, Malysa A, Wieckiewicz M. Effects of Polishing and Artificial Aging on Mechanical Properties of Dental LT Clear® Resin. J Funct Biomater. 2023 May 25;14(6):295. doi: 10.3390/jfb14060295.
Altaie SF. Tribological, microhardness and color stability properties of a heat-cured acrylic resin denture base after reinforcement with different types of nanofiller particles. Dent Med Probl. 2023;60(2):295–302. doi:10.17219/dmp/137611
3. Please clarify why did you use 20 samples in one group? Did you perform sample power and size calculation or other reason determine the number?
4. Figure 1 and Figure 4 and Figure 6 are unclear. Please provide pictures with high resolution.
5. The title, aim of the study and conclusions have to correspond one to each other. Please revise this issue in the abstract and manuscript body.
6. The Authors have to clearly write in the conclusions whether the properties of the 3D resin are the same, better or worse than traditional PMMA resin.
The language of the manuscript looks fine.
Author Response
The manuscript presents interesting study. However I found a few major flaws:
Point 1:
- Authors have to rationale the study and write why this study is novel and important. Please add a paragraph in Introduction before the aim of the study.
Response 1:
We have restructured the abstract and added the necessary information in the introduction. Thank you!
Point 2:
- Authors cited only 14 references (references should have at least 25 items)!Almost half of them are older than 10 years.Authors should remove all references older than 10 years and consider to cite current and reliable literature related to the topic (citation of mentioned literature is voluntarily):
Paradowska-Stolarz A, Malysa A, Mikulewicz M. Comparison of the Compression and Tensile Modulus of Two Chosen Resins Used in Dentistry for 3D Printing. Materials (Basel). 2022 Dec 15;15(24):8956. doi: 10.3390/ma15248956.
Paradowska-Stolarz A, Wezgowiec J, Mikulewicz M. Comparison of Two Chosen 3D Printing Resins Designed for Orthodontic Use: An In Vitro Study. Materials (Basel). 2023 Mar 10;16(6):2237. doi: 10.3390/ma16062237.
Paradowska-Stolarz AM, Wieckiewicz M, Mikulewicz M, et al. Comparison of the tensile modulus of three 3D-printable materials used in dentistry [published online as ahead of print on May 25, 2023]. Dent Med Probl. doi:10.17219/dmp/166070
Paradowska-Stolarz A, Wezgowiec J, Malysa A, Wieckiewicz M. Effects of Polishing and Artificial Aging on Mechanical Properties of Dental LT Clear® Resin. J Funct Biomater. 2023 May 25;14(6):295. doi: 10.3390/jfb14060295.
Altaie SF. Tribological, microhardness and color stability properties of a heat-cured acrylic resin denture base after reinforcement with different types of nanofiller particles. Dent Med Probl. 2023;60(2):295–302. doi:10.17219/dmp/137611
Response 2:
Thank you very much for the suggestions. We have incorporated some of the suggested references. Than you!
Point 3:
- Please clarify why did you use 20 samples in one group? Did you perform sample power and size calculation or other reason determine the number?
Response 3:
Thank you very much for your feedback. Due to the constrains on the programming of the testing and the economic factors involved we could not include more than 20 samples per group. Sample power size calculation was not considered due to the nature of the other constrains.
Point 4:
- Figure 1 and Figure 4 and Figure 6 are unclear. Please provide pictures with high resolution.
Response 4:
Thank you very much. We enhanced as much as technically possible the images mentioned above.
Point 5:
- The title, aim of the study and conclusions have to correspond one to each other. Please revise this issue in the abstract and manuscript body.
Response 5:
Thank you for your suggestion. We modified the text accordingly!
Point 6:
- The Authors have to clearly write in the conclusions whether the properties of the 3D resin are the same, better or worse than traditional PMMA resin.
Response 6:
We modified the conclusion as requested. Thank you very much.
Reviewer 4 Report
The article “Innovative new orthodontic occlusal splints manufacturing using triple jetting technology: Comparison with polymethyl- methacrylate (PMMA) of its mechanical properties.” aims to determine the mechanical proprieties of the 3D printing material and to compare it to the conventional PMMA.
Abstract:
lines 25-7
“The control material has a statistically significant Young’s Modulus of Compression and Tensile Strength (p<0.05).”
Statistically significant higher? Lower?
Please specify
Lines 28-29:
“Maximum Bending Stress at Maximum Load (MPa) is statistically more significant for the 3d printed samples.”
Please rephrase using “higher or lower values”.
Lines 64-8:
Please declare whether PMMA salt and pepper samples were cured under vacuum or not
Lines 286-88:
“Regarding the compression, our results concluded that the control material (PMMA) has a statistically significant Young’s Modulus of Compression and Tensile Strength (p<0.05).”
Also here please specify if Young’s Modulus of Compression was significantly higher or lower.
The authors could elaborate in a paragraph the evidence obtained from the SEM images. Cohesion, transition zones, at different magnifications and relate them to the mechanical properties investigated.
The authors used the word “orthodontic” in the title.
No further references were made elsewhere in the document. The reviewer thinks that the triple jetting technology could also be beneficial for post-orthodontic retainers.
The authors are encouraged to add the following sentence at the end of the discussion before the conclusion (this could also support what was stated by the authors: “can replace in many applications the classical heat-cured resin”):
“Further studies should therefore be performed to investigate whether triple jetting technology could be suitable for occlusal retainers after conventional (cite PMID: https://pubmed.ncbi.nlm.nih.gov/28778722/), lingual (cite PMID: https://pubmed.ncbi.nlm.nih.gov/19350058/) or aligner-base orthodontic treatments (cite PMID: https://pubmed.ncbi.nlm.nih.gov/34965910/)”
Please add limitations of the current study.
Please add future studies to be performed after considering the current study’s findings.
Author Response
The article “Innovative new orthodontic occlusal splints manufacturing using triple jetting technology: Comparison with polymethyl- methacrylate (PMMA) of its mechanical properties.” aims to determine the mechanical proprieties of the 3D printing material and to compare it to the conventional PMMA.
Point 1:
Abstract:
lines 25-7
“The control material has a statistically significant Young’s Modulus of Compression and Tensile Strength (p<0.05).”
Statistically significant higher? Lower?
Please specify
Response 1:
Thank you for your feedback. We added that is statistically significant higher.
Point 2:
Lines 28-29:
“Maximum Bending Stress at Maximum Load (MPa) is statistically more significant for the 3d printed samples.”
Please rephrase using “higher or lower values”.
Response 2:
Thank you for your feedback. We added that is statistically significant higher.
Point 3:
Lines 64-8:
Please declare whether PMMA salt and pepper samples were cured under vacuum or not
Response 3:
According to the manufacturer’s instructions the samples were cured in vacuum. We added the text also in the manuscript.
Point 4:
Lines 286-88:
“Regarding the compression, our results concluded that the control material (PMMA) has a statistically significant Young’s Modulus of Compression and Tensile Strength (p<0.05).”
Also here please specify if Young’s Modulus of Compression was significantly higher or lower.
Response 4:
We have also added the necessary details to the manuscript. Thank you.
Point 5:
The authors could elaborate in a paragraph the evidence obtained from the SEM images. Cohesion, transition zones, at different magnifications and relate them to the mechanical properties investigated.
Response 5:
Thank you very much for your feedback. We modified the section 3.4 and added the necessary discussions regarding the SEM analysis to the aforementioned section.
Point 6:
The authors used the word “orthodontic” in the title.
No further references were made elsewhere in the document. The reviewer thinks that the triple jetting technology could also be beneficial for post-orthodontic retainers.
Response 6:
Thank you very much for your remark. Yes, it could be used in various applications especially in the orthodontic field. We believe that there is good amount of evidence that the material printed using the polyjet technology achieves better results, better mechanical results, more predictability of the material and better ease of use for the technician-doctor-patient combo.
Point 7:
The authors are encouraged to add the following sentence at the end of the discussion before the conclusion (this could also support what was stated by the authors: “can replace in many applications the classical heat-cured resin”):
“Further studies should therefore be performed to investigate whether triple jetting technology could be suitable for occlusal retainers after conventional (cite PMID: https://pubmed.ncbi.nlm.nih.gov/28778722/), lingual (cite PMID: https://pubmed.ncbi.nlm.nih.gov/19350058/) or aligner-base orthodontic treatments (cite PMID: https://pubmed.ncbi.nlm.nih.gov/34965910/)”
Response 7:
Thank you for your suggestion. We added the phrase and the citations. Thank you!
Point 8:
Please add limitations of the current study.
Response 8:
Limitations added to the discussion section. Thank you.
Point 9:
Please add future studies to be performed after considering the current study’s findings.
Point 9:
Thank you very much! We have added this to the discussion section.
Round 2
Reviewer 1 Report
Please, go through the pdf file "biomedicines-2493594 - Review round _1 - Authors response - Reviewer response".
Please, go through the pdf file sent in the first round, again.

English writing in the manuscript is improved.
Author Response
Point 1:
- Title should be rewritten. It should be written clear and concise, not as a sentence. Response 1:
Thank you very much for your suggestion. We have modified the title as suggested. Reviewer response 1:
New title is “Comparing Mechanical Properties of Triple Jetting Technology and PMMA in Orthodontic Occlusal Splint Manufacturing”. It seems that there is misunderstanding, again. Mechanical properties can be compared between parts produced with two different technologies, or parts of two different materials. It is not possible to compare mechanical properties between technology and material.
Please, rewrite.
Response 1: Thank you for clearing the misunderstanding. You are very right we have corrected the title accordingly!
Point 2:
- Abstract should be rewritten, especially paying attention on technical English expressions. Concept of the abstract should be changed, because there is one “introduction” sentence and then continuing with not well written goal pf research.
o Please, rewrite the first Abstract sentence (lines 18,19). After this sentence there should be at least one more introducing the 3D printing and its influence in this research. With that, good "connection" should be made with the rest of the abstract.
Response 2:
Thank you very much for your suggestions. We modified the abstract as suggested and we believe that it made it clearer and more concise. Thank you!
Reviewer response 2:
Sentence is not added after the first one in order to better introduce and connect. Expanding the first sentence is not enough to make the introduction.
Line 29 – Wherever I could, I highlighted the term “traction” (somewhere I missed) as not proper one for tensile testing and characteristics. I did not highlight this one in the abstract (but I did in previous abstract sentences), authors did not change it. Please, go through the text and make the terms uniform, i.e., “traction” should be replaced with adequate term.
Response 2:
Thank you. We have added a linking sentence in the abstract to better connect the first phrase with the rest of the abstract.
We also have changed the traction with “tensile testing”.
Point 3:
- Keywords should be arranged in a different order.
Response 3:
Thank you. We have arranged the keyword to the importance that they have to the article.
Reviewer response 3:
All the keywords must be present both in the title and abstract. You are pointing out the mechanical properties in your manuscript, so, they should be placed in the keywords.
Response 3:
Thank you. We have added also compression testing; 3 point bending testing; to the keywords list.
Point 4:
- Lines 41,42 - Reference 3 is not appropriate, concerning the fact that no polymerization is mentioned in the referenced manuscript.
Response 4:
Thank you for your remark. We modified the paragraph to be appropriate.
Reviewer response 4:
Authors changed the paragraph a little, but reference 3 is still present and not changed. And, once again, reference is not appropriate, concerning the fact that no polymerization is mentioned in the referenced manuscript.
Response 4:
We changed it as suggested. Thank you!
Point 5:
- Lines 43,44 - This sentence is quite confusing. These techniques are completely opposite. Subtractive is conventional, well known (“old”) manufacturing approach, and because of that it is not known from recent period. Additive (3D) printing approach is newer, and in comparison with subtractive it can be said that it appeared recently.
Response 5:
Thank you for your careful reading of our manuscript. We have modified the sentences to be accurate. Reviewer response 5:
Alright.
Point 6:
- Line 46 – Reference 4 is also confusing.
Response 6:
We modified the reference to better fit the content. Thank you!
Reviewer response 6:
Alright.
Point 7:
- Line 4,48 – This sentence (also in the abstract) should be rewritten. Triple jetting technology is advanced part of polyjet technology, so it should be placed in brackets. Also, not only these techniques are represented in dentistry 3D printing.
Response 7:
Thank you for your feedback! We have rearranged the paragraph according to your suggestions. Thank you!
Reviewer response 7:
Alright. SLS is also present in dentistry.
Response 7:
Also added SLS with explanation.
Point 8:
- Lines 47-55 - SLA is described, but where are the other two, in previous sentence, mentioned 3D printing techniques applied in dentistry?
Response 8:
Thank you. We did not want to exhaustively explain all the technologies involved, but just the one which we believe is newer and innovative (polyjet).
Reviewer response 8:
If you explain one technology (SLA), which is not present in this research methodology, than you should shortly explain all of them mentioned. Polyjet (triple jetting) should be explained in more details, as the one used in this research.
Response 8:
We added more explanations to the polyjet technology. Thank you!
Point 9:
- Introduction is too short, it does not give any important, or even relevant information regarding the current situation in the field which authors want to give their scientific contribution. It is not introducing reader into what can be expected on proper manner. 5 references is not enough.
Response 9:
We have rearranged the entire paragraph to better suit the needs of the reader and also added better references.
Reviewer response 9:
Lines 73 and 89 – these kind of underlined sentences are too colloquial for the manuscript in the journal of this category.
Lines 73 to 78 – pointed out finding is not properly described. “Conventionally cured samples” from which material? ASIGA is the resin material?
Lines 79 to 82 – What is “CAD group”?
Response 9:
We corrected the paragraphs. CAD – computer aided design; Asiga is the manufacturer of 3d printers and resins.
Point 10:
- Figure 1 is not clear et al.
Response 10:
We rearranged figure 1 to be a bit more clear.
Reviewer response 10:
Authors widen the figure 1 and made it even more unclear, with disturbed resolution. The point was not to make it wider, but to change it with different figure in order to, actually, be able to see anything on it.
Figures for flexure and tensile testing are alright, but, figure for compression testing is not proper, because grips of the machine are not visible, nor the sample.
Response 10:
We tried to enlarge figure 1, unfortunately this is the best we can get at this point.
We modified so the grips can be visible. Thank you!
Point 11:
- Line 94 – Three point bending test.
Response 11:
Thank you very much for your observation. We have corrected it accordingly!
Reviewer response 11:
Alright. All the terms throughout the manuscript should be unified and technically appropriate.
Response 11:
Thank you!
Point 12:
- Please, use regular diagrams for mechanical testing’s which are generated by the testing machine. Not the cloud charts.
Response 12:
We have added the descriptive charts as requested. Thank you.
Reviewer response 12:
I did not ask for descriptive charts, but I made remark to show the Stress-Strain diagrams (curves) for each mechanical testing. Cloud charts are not proper form of presenting the data regarding the mechanical testing’s, and they should be removed. Charts/plots which you newly added are for statistical explanation, and they can stay.
Response 12:
Unfortunately, at this point of time these charts cannot be exported anymore. The data was processed, and these were not saved. We removed the cloud charts.
Thank you!
Point 13:
- It is necessary to use technical terminology for mechanical testing results. Response 13:
Thank you, we modified the terms accordingly.
The same as Reviewer response 12.
Response 13:
Same as 12.
Point 14:
- All the underlined text in the discussion should be rewritten, re-conceptualized, and placed in the introduction. Leaving almost no discussion.
Response 14:
Thank you very much for the feedback and the suggestions. We rephrased the discussion and moved it to the introduction section.
We have added new discussion with appropriate references.
Reviewer response 14:
Alright.
Point 15:
- Conclusion is weak.
Response 15:
We rephrased the conclusion. Thank you!
Reviewer response 15:
Alright.
Point 16:
- 14 references in total regarding this subject is too small.
Response 16:
We have rearranged the manuscript and enhanced it. We further added important references in both introduction and discussions sections.
Reviewer response 16:
Introduction is improved, but it should be additionally improved with pointing out of the previous researches in more presenting way.
Thank you! We added the suggested phrase at the beginning of the introduction!
Discussion is alright.
Additional comments
Please observe, in complete, the pdf document with highlighted and underlined text attached for the first revision. Please read all the remarks.
Line 122 – Concerning the fact that you shortly explained (and it should be explained even more) each 3D printing process used in dentistry, “salt and pepper” process should be also explained shortly.
The technique has been added and described.
Lines 197-199 - I made remark to rewrite this sentence, but it is not done. Line 219 – What is T-test?
The T-test can be sometimes found as student’s T test. We rewrote the sentence.
Table 4 caption is not proper. Remark from the pdf is not followed. I highlighted the text because terminology should be changed.
We modified it accordingly.
Figures 10 and 11 should be placed in between the text.
We modified it accordingly!
There are no measured values for the Engineered Stress (MPa) and Engineered Strain (%). That is leading to the conclusion that point of the paper is in statistical comparison of mechanical characteristics between triple jetting 3D printed and conventionally manufactured PMMA samples, and tone of the title and explanations should be pointed out in that manner.
Thank you for your kind remark. We modified the title to best fit the contents of the paper.
Reviewer 2 Report
No more comments
Minor English editing is required
Author Response
Thank you very much!
Reviewer 3 Report
The manuscript has been revised correctly. I don't have further remarks.
I don't have major remarks only minor language mistakes.
Author Response
Thank you very much!
Reviewer 4 Report
The authors have improved the manuscript according to the reviewer's suggestions
Author Response
Thank you very much!
Round 3
Reviewer 1 Report
· You used biocompatible clear 131 MED610 dental resin. Why did you wrote “composite” in the title?
“Statistical comparison of the mechanical Properties of a 3D printed composite dental resin through Triple Jetting Technology and a conventional polymethylmethacrylate PMMA resin in Orthodontic Occlusal Splint Manufacturing “
· There is no need to literally add all the words in keywords.
Keywords: polymethylmethacrylate (PMMA); 3d printing; polyjet; triple jetting technology; mechanical properties; occlusal splint; tensile testing; compression testing; 3 point bending testing scanning electron microscopy (SEM);
· Lines 86-88 – This new paragraph is not necessary. It should be removed.
· Line 131 – “…biocompatible clear 131 MED610…” – the term “resin” is missing.
· For Figure 1. Please add sample figure on the left, and make it the same as figures 2 and 3. To have the same size and look.
Just minor changes.
Author Response
- You used biocompatible clear 131 MED610 dental resin. Why did you wrote “composite” in the title?
Yes you are very right. We rearranged the title. Thank you.
“Statistical comparison of the mechanical Properties of a 3D printed composite dental resin through Triple Jetting Technology and a conventional polymethylmethacrylate PMMA resin in Orthodontic Occlusal Splint Manufacturing “
- There is no need to literally add all the words in keywords.
Keywords: polymethylmethacrylate (PMMA); 3d printing; polyjet; triple jetting technology; mechanical properties; occlusal splint; tensile testing; compression testing; 3 point bending testing scanning electron microscopy (SEM);
We deleted the unnecessary keywords in the list.
- Lines 86-88 – This new paragraph is not necessary. It should be removed.
We removed it. Thank you.
- Line 131 – “…biocompatible clear 131 MED610…” – the term “resin” is missing.
We added the term "resin" to the phrase.
- For Figure 1. Please add sample figure on the left, and make it the same as figures 2 and 3. To have the same size and look.
We added the sample photography for the compression test and rearranged them accordingly. Thank you!